# Physical Properties of Carboxymethyl Cellulose from Palm Bunch and Bagasse Agricultural Wastes: Effect of Delignification with Hydrogen Peroxide

**DOI:** 10.3390/polym12071505

**Published:** 2020-07-07

**Authors:** Rungsiri Suriyatem, Nichaya Noikang, Tamolwan Kankam, Kittisak Jantanasakulwong, Noppol Leksawasdi, Yuthana Phimolsiripol, Chayatip Insomphun, Phisit Seesuriyachan, Thanongsak Chaiyaso, Pensak Jantrawut, Sarana Rose Sommano, Thi Minh Phuong Ngo, Pornchai Rachtanapun

**Affiliations:** 1Division of Cosmetic Science and Health Products, School of Pharmacy, Eastern Asia University, Pathum Thani 12110, Thailand; rungsiri@eau.ac.th; 2School of Agro-Industry, Faculty of Agro-Industry, Chiang Mai University, Chiang Mai 50100, Thailand; nichaya.aong@gmail.com (N.N.); tamolwan.aumaim@gmail.com (T.K.); jantanasakulwong.k@gmail.com (K.J.); noppol@hotmail.com (N.L.); yuthana.p@cmu.ac.th (Y.P.); chayatip@yahoo.com (C.I.); phisit.s@cmu.ac.th (P.S.); thachaiyaso@hotmail.com (T.C.); 3The cluster of Agro Bio-Circular-Green Industry (Agro BCG), Chiang Mai University, Chiang Mai 50100, Thailand; pensak.amuamu@gmail.com (P.J.); sarana.s@cmu.ac.th (S.R.S.); 4Center of Excellence in Materials Science and Technology, Chiang Mai University, Chiang Mai 50200, Thailand; 5Department of Pharmaceutical Sciences, Faculty of Pharmacy, Chiang Mai University, Chiang Mai 50200, Thailand; 6Plant Bioactive Compound Laboratory (BAC), Department of Plant and Soil Sciences, Faculty of Agriculture, Chiang Mai University, Chiang Mai 50200, Thailand; 7Department of Chemical Technology and Environment, The University of Danang-University of Technology and Education, Danang 550000, Vietnam; ntmphuong@ute.udn.vn

**Keywords:** bleaching, carboxymethylation, cellulose, CMC film, lignin, viscosity

## Abstract

The aim of this work was to synthesize carboxymethyl cellulose (CMC) and produce CMC films from the cellulose of palm bunch and bagasse agricultural waste. The effect of various amounts of H_2_O_2_ (0–40% *v*/*v*) during delignification on the properties of cellulose, CMC, and CMC films was studied. As the H_2_O_2_ content increased, yield and the lignin content of the cellulose from palm bunch and bagasse decreased, whereas lightness (*L**) and whiteness index (WI) increased. FTIR confirmed the substitution of a carboxymethyl group on the cellulose structure. A higher degree of substitution of CMC from both sources was found when 20%–30% H_2_O_2_ was employed. The trend in the *L** and WI values of each CMC and CMC film was related to those values in their respective cellulose. Bleaching each cellulose with 20% H_2_O_2_ provided the cellulose with the highest viscosity and the CMC films with the greatest mechanical (higher tensile strength and elongation at break) and soluble attributes, but the lowest water vapor barrier. This evidence indicates that cellulose delignification with H_2_O_2_ has a strong effect on the appearance and physical properties of both CMCs.

## 1. Introduction

Carboxymethyl cellulose (CMC) is one of the most important cellulose derivatives. A large amount of commercial CMC is used in various applications such as ceramic foam [1], paper [2], textiles [3], pharmaceutics [4], food [5], and biodegradable films [6,7]. CMC is derived from cellulose, which is non-toxic to human health. Cellulose is commonly found in plant cell walls. It is usually conjunct with hemicellulose and lignin. The variety of plant affects the composition and the cellulose content. The other substances, especially lignin, also make it difficult to obtain pure cellulose. To get rid of lignin, numerous oxidizing agents are used in most lignocellulosic materials for bleaching [8,9,10,11]. They are categorized as chlorine compounds and non-chlorine compounds. Among all of them, there are just a few eco-friendly reagents. Hydrogen peroxide (H_2_O_2_) is one of the commercial reagents used in environmentally friendly and energy-saving bleaching systems [9,12]. Different cellulose sources have been reported to implement H_2_O_2_ into the bleaching treatment, such as papaya peel [12], durian rind [9], oil palm empty-fruit-bunch [13], and cotton [14]. In addition, some studies have reported the influence of oxidizing agent concentration on the properties of the final products of various cellulose sources such as pure cellulose and paper [15], olive tree branch [16], and rice husk [17].

Nowadays, sustainable and ecological materials are the subject of global focus with constant growing interest. Agricultural waste is a cellulose source which is cheap, various and abundant. In literature, many research has studied the synthesis of CMC from agricultural waste cellulose sources. The examples include CMC from sugar beet pulp [18], cavendish banana pseudostem [10], cotton linters [19], sago waste [20], papaya peel [12], mulberry paper [21], corn husk [22], *Mimosa pigra* peel [23], and durian rind [9,24]. 

Until today, there has not been any research on the effect of delignification with H_2_O_2_ on the physical properties of cellulose and CMC from palm bunch and bagasse agricultural wastes, including their CMC films. Therefore, the present study aimed to demonstrate the influence of bleaching cellulose with H_2_O_2_ on the properties of cellulose, CMC and CMC films from palm bunch and bagasse. The cellulose and lignin contents of the raw materials were determined. The percent yield, lignin content, optical properties, morphology, and chemical structure of the bleached cellulose was evaluated. In addition, the percent yield, degree of substitution, optical properties, viscosity, morphology, and chemical structure of the CMCs were investigated. Moreover, the optical and mechanical properties, solubility, and water vapor transmission rate of the CMC films were investigated.

## 2. Materials and Methods 

### 2.1. Materials

Palm bunch was collected from the palm garden in Chiang Mai Floraville Village (Chiang Mai, Thailand). Bagasse was obtained from a local sugarcane juice shop at Central Plaza Chiang Mai Airport (Chiang Mai, Thailand). Sodium hydroxide and glacial acetic acid were purchased from Merck (Darmstadt, Germany). Monochloroacetic acid was purchased from Sigma Aldrich (Steinhiem, Germany). Hydrogen peroxide, glycerol, ethanol, and methanol were purchased from Northern Chemicals and Glasswares Ltd., Part. (Chiang Mai, Thailand). All reagents were of analytical grade and used as received.

### 2.2. Sample Preparation

Palm bunch and bagasse were cut into pieces of 20 mm length, sun-dried for around two days and further dried at 60 °C in an oven for 24 h. The dried samples were ground into powder using a hammer mill (Armfield, England). The cellulose and lignin contents of the raw materials were determined according to TAPPI (Technical Association of the Pulp and Paper Industry) standard methods T 203 cm-09 [25] and T 222 om-15 [26], respectively. Shortly afterwards, sample powder (3 g), distilled water (160 mL), glacial acetic acid (0.5 mL), and sodium chloride (1.5 g) were mixed in a beaker and stirred continuously for 1 h at 80 °C. The beaker was cooled to 10 °C, filtered, and washed with acetone. The product was dried in an oven for 24 h at 80 °C. Cellulose content (%) was calculated by multiplying the ratio of product weight/sample weight by 100. In order to determine lignin content, in brief, sample powder (1 g) was placed in a beaker at 2 °C. Cold (10–15 °C) sulfuric acid (72%, 15 mL) was gradually introduced, and the mixture was continuously stirred for 2 h. Distilled water (560 mL) was added. The mixture was boiled for 4 h and left standing to allow precipitation overnight. The mixture was filtered, washed with water three times, and dried in an oven for 6 h at 105 °C. The final product was later weighed. The lignin content (%) was calculated by multiplying the ratio of product weight/sample weight by 100.

### 2.3. Cellulose Extraction and Bleaching

Cellulose from palm bunch and bagasse was extracted and bleached according to the method described by Rachtanapun et al. [9]. In brief, the pulp samples (10 g) were mixed with 30% *w*/*v* NaOH (100 mL) in a beaker. The mixture was heated to 100 °C, stirred continuously for 3 h and filtered. The solid phase was rinsed with distilled water several times until the remaining water was clear. The cellulose pulp was dried in an oven at 55 °C for 24 h. The dried pulp (10 g) was subjected to bleach with various concentrations of H_2_O_2_ solution (0%, 10%, 20%, 30%, and 40% *v*/*v*) in a hot water bath at 80 °C for 2 h. Finally, the bleached pulp was rinsed with distilled water to remove the residual lignin and dried in an oven at 55 °C for 24 h. The percent yield of the cellulose products was calculated using Equation (1): (1)$$\mathrm{Yield}\;\mathrm{of}\;\mathrm{cellulose}\;\left(\%\right)=\frac{W_{a}}{W_{b}}\times 100$$
where *W_a_* and *W_b_* were the weight of cellulose after and before bleaching, respectively. The Kappa number (*K*) was applied in this work according to TAPPI standard method T 236 om-99 [27] to determine the bleaching ability or degree of delignification of the pulp. Briefly, the pulp (3 g) was suspended in distilled water (500 mL) with continuous stirring for 5 min at 25 °C. The addition of 0.1 M potassium permanganate (100 mL) and 4 M sulfuric acid (100 mL) was later performed, and the mixture was then stirred continuously for 5 min. Distilled water was added to achieve a total volume of 1.0 L, and the mixture was constantly stirred for 10 min. The cessation of the reaction was achieved with the addition of 1 M potassium iodide (20 mL). Immediately, the free iodine in the mixture was titrated with 0.2 M sodium thiosulfate. A few drops of starch solution were used as an indicator. The K value was calculated using the equation described elsewhere [27]. The approximate lignin level (%) of the bleached cellulose samples was calculated using Equation (2) [27]:(2)Lignin (%)=K×0.13

### 2.4. Synthesis of CMC

Before synthesis, the bleached cellulose from palm bunch and bagasse was ground into a fine powder (under 60 mesh) using an Ultra centrifugal mill ZM 200 (RETSCH, Germany). The cellulose was converted to CMC according to the method described by Rachtanapun et al. [9]. Cellulose (15 g), 40% *w*/*v* NaOH (50 mL) and isopropanol (IPA, 450 mL) were mixed and continuously stirred in a beaker at 50 °C for 1 h. The mixture was gradually supplemented with the solution of chloroacetic acid/IPA (18 g: 18 mL), further stirred for 30 min and placed in an oven at 55 °C for 3.5 h. The liquid phase of the mixture was removed. The solid phase was mixed with methanol (225 mL) and neutralized with glacial acetic acid. The mixture was filtered and rinsed with 70% *v*/*v* ethanol (225 mL) 5 times and a final time with 95% *v*/*v* methanol (225 mL). The last product, CMC, was dried in an oven at 55 °C for 12 h and kept in a polyethylene bag until being used. The percent yield of the CMC was calculated using Equation (3):(3)$$\mathrm{Yield}\;\mathrm{of}\;\mathrm{CMC}\;\left(\%\right)=\frac{W_{CMC}}{W_{c}}\times 100$$
where *W_CMC_* was the weight of synthesized CMC; *W_c_* was the weight of cellulose. The degree of substitution (DS) of the CMC was measured using the USP XXXII method described for croscarmellose sodium, the procedure of which was described elsewhere [23].

### 2.5. Characterization of Cellulose and CMC Powder 

The lightness (*L**) value of the cellulose and CMC samples was measured using a colorimeter (CR-400, Konica Minolta, Shanghai, China). Sample powder (2 g) was set in a sample holder placed above the instrument. The testing button was pressed, and the *L** value was shown on the result screen. Each test was run in triplicate. The whiteness index (*WI*) value was calculated using a method described elsewhere by the authors [28].

Morphological investigation of the cellulose and CMC samples was performed using a scanning electron microscope, SEM (Quanta 200 3D, FEI, Hillsboro, OR, USA). The samples were adhered to carbon tape containing a specimen stub and sputter-coated with a thin layer of gold using a sputter coater (SPI-Module, Wilmington, MA, USA). The observation was run on a voltage of 15 kV with 1000× magnification.

An FTIR spectrometer (Tensor 27, Bruker, Ettlingen, Germany) was used to provide transmission infrared spectra of the samples. A pellet of each sample was mounted directly in the pellet holder. The test was run in the range of 4000–400 cm^−1^, with a resolution of 4 cm^−1^. Before the test, pellets of the samples were prepared by mixing and pressing the samples (~2 mg) with KBr.

A Rapid Visco Analyzer (RVA-4, Newport Scientific, Jessup, MD, USA) was used to measure the viscosity of the CMC samples. CMC (3 g) was dissolved in distilled water (25 mL) at 80 °C for 10 min. The CMC solution was poured into the test cup and placed in the sample holder. Initially, the speed of testing was set at 960 rpm, for 10 s at 30 °C. Afterwards, the temperature was varied (30, 40, and 50 °C) for 6 min with a speed of 160 rpm. All measurements were run in triplicate.

### 2.6. Preparation of CMC Films

To prepare the CMC film, CMC (3 g), distilled water (100 mL), and glycerol (0.9 g) were mixed in a beaker, heated to 80 °C, and continuously stirred for 10 min. The mixture was cooled to 25 °C, cast on an acrylic plate, and dried in an oven at 40 °C for 24 h. The dried film was peeled and kept at 25 °C, 52 ± 1 %RH (relative humidity) in a desiccator until testing. The thickness of the films was measured using a micrometer model GT-313-A (Gotech Testing Machine Inc., Taichung, Taiwan).

### 2.7. Properties of CMC Films

The *L** and WI of the film samples were measured using the same method as described above in Section 2.5. The test was run in triplicate.

Tensile properties (tensile strength, TS, and elongation at break, EB) were determined according to ASTM (American Standard for Testing and Materials) D882-12 [29]. The film was cut into ten rectangular specimens (15 mm × 150 mm) and tested using a Universal Testing Machine Model 1000 (Instron corp., Canton, MA, USA). The initial grip separation and the speed of the test were set at 100 mm and 10 mm/min, respectively. 

To measure water solubility, the film was cut into a square shape (20 mm × 20 mm). The film specimen was soaked in distilled water (50 mL) and shaken continuously for 2 h. The residue was filtered and dried in an oven at 105 °C for 24 h. The test was applied in triplicate. The solubility of the film samples was calculated using Equation (4):(4)$$\mathrm{Solubility}\;\left(\%\right)=\frac{W_{1}-W_{2}}{W_{1}}\times 100$$
where *W*_1_ was the weight of the sample before testing; *W*_2_ was the weight of the dried residual.

The water vapor transmission rate (WVTR) of the CMC films was measured according to ASTM-E96/E9M-16 [30] using a cup method. Each CMC film sample was prepared by cutting it into a circle (Ø ≈ 8 cm). The test cup was filled with dried silica gel (10 g), covered with a film sample and sealed with paraffin wax. The cup was weighted and placed in a desiccator at 25 °C, 52 ± 1 %RH. The cup was weighted every 24 h for 7 days. The WVTR of the films was calculated using Equation (5):(5)WVTR=slope/A
where *slope* was obtained from the linear curve of the plot between weight gain (y axis) and time (x axis); *A* was the area of the specimen.

### 2.8. Statistical Analysis

Data were analyzed using SPSS software (Version 11, SPSS Inc., Chicago, IL, USA). One-way analysis of variance (ANOVA) and Duncan’s multiple range test (*p* ≤ 0.05) were carried out. 

## 3. Results

The cellulose and lignin contents of the raw materials were determined. It was found that palm bunch had 29.3% cellulose and 41.5% lignin, while bagasse had 42.5% cellulose and 23.8% lignin. These values were similar to the results of other research where empty palm fruit bunches were found to contain 27.2% cellulose and 43.2% lignin [31]. Phinichka and Kaenthong [32] reported that sugarcane bagasse had 41.4% cellulose and 19.0% lignin. Also, the lignin contents of sugarcane bagasse were found to be 25%–28% by Sakdaronnarong et al. [33] and 17%–24% by Masarin et al. [34]. The variety of the chemical composition of original wood samples with the same part and tree type depends on soil conditions, geographic location and climate [35]. 

The code and appearance of cellulose and CMC from palm bunch and bagasse are listed in Table 1. The letters “p” and “b” shown after a code imply the maternal materials palm bunch and bagasse, respectively. 

### 3.1. Characterization of Cellulose

In this work, the percent yield of cellulose from palm bunch and bagasse gradually declined when H_2_O_2_ concentration increased (Figure 1a). Palm bunch cellulose loses its weight more than bagasse cellulose after bleaching. This may be because the cellulose of palm bunch has higher lignin content than that of bagasse. The Kappa number (*K*) was used to indicate the remaining lignin content, implying bleaching ability. Figure 1b,c shows that the trend in *K* value and lignin content of the bleached cellulose was similar, and this was also related to the trend in percent yield. A similar observation was found with sweet bamboo Kraft pulp by Kamthai and Puthson [36]. The decrease in the yield of cellulose after bleaching may be caused by a decrease in lignin and hemicellulose content [37]. However, at higher levels of H_2_O_2_ concentration or lower pH, the decrease in the yield of cellulose may be because of the depolymerization of cellulose macromolecules to shorter molecules, which are removed more easily from the fiber surface. This observation can be confirmed and further discussed using the SEM results.

The lightness (*L**) and whiteness index (WI) were used as factors affecting the bleaching treatment of the cellulose. These increased with increasing H_2_O_2_ concentrations (Figure 1d,e). This may be because of the bleaching effect of the perhydroxyl anion (HO_2_^−^) [38]. In this study, H_2_O_2_ in aqueous solution was an equilibrium with the HO_2_^−^ and the peroxo dianion (O_2_^2−^). In view of the bleaching mechanism, the perhydroxyl anion may generate perhydroxyl radicals (HO_2_^•^) and further dissociate to create the radical anion (O_2_^−^^•^), superoxide. The superoxide is considered one of the active bleaching agents [39].

### 3.2. Characterization of CMC 

CMC from palm bunch (CMCp) and bagasse (CMCb) was synthesized from their cellulose bleached with various H_2_O_2_ concentrations (0%–40%). The percent yield of CMCp ranged from 120.52% to 135.78%, and that of CMCb ranged from 135.52% to 142.41% (Figure 2a). At the beginning, the percent yield of the CMCs increased with increasing H_2_O_2_ concentration (0%–20%). Nevertheless, it decreased when H_2_O_2_ concentration was over 20%. The highest yield was given by the CMC from 20% H_2_O_2_ -bleached cellulose from both maternal materials. The degree of substitution (DS) of the CMCs showed that the trend was related to percent yield (Figure 2b). The *L** and WI values of each CMC were related to its cellulose as a primary material (Figure 2c,d). The DS value of CMCp and CMCb ranged between 0.17–0.36 and 0.23–0.53, respectively. The DS was used to indicate the solubility of the CMC. DS values between 0.0 and 0.4 mean that CMC is insoluble but swellable, and above this range mean that the CMC is completely water-soluble [18]. The highest DS value was found in CMCp and CMCb made from their respective cellulose and bleached with 20% H_2_O_2_. Lignocellulosic material is generally composed of cellulose, hemicellulose, and lignin connecting assembly. Lignocellulosic material is generally composed of connected cellulose, hemicellulose, and lignin. Following bleaching with higher H_2_O_2_ concentrations, the cellulose may be more available. Thus, the reaction between the cellulose chain and etherifying agent to generate the CMC may be enhanced. However, at too high a concentration of H_2_O_2_, the cellulose may be depolymerized, which affects the DS decline. Other research has made similar observations to our result, such as work on CMC from *Mimosa pigra* peel [23], CMC from durian rind [9], and CMC from the waste of cotton ginning [40].

It has been found that the viscosity of CMC could be influenced by many factors, such as CMC content [41], sodium hydroxide concentration [9,23], and temperature [9,23]. This work focused on the effect of H_2_O_2_ concentration and temperature on the viscosity of CMCp and CMCb. Figure 3a,b presents the viscosity of CMCp and CMCb, respectively, at 30, 40, and 50 °C. The result shows that the viscosity of each type of CMCp and CMCb was lower when the temperature was higher. This is because the higher temperature could enhance the rate of molecular interchange, at the same time as decreasing the cohesive forces, reflecting the lower viscosity. A similar relationship was found for CMC from durian rind [9] and CMC from *Mimosa pigra* peel [23]. At the same temperature, the trend in viscosity for CMCp and CMCb followed the same pattern. It increased with increasing H_2_O_2_ concentrations of up to 20% and decreased after this point (30%–40% H_2_O_2_), agreeing with the tendency of the DS results mentioned previously. It seems that the viscosity of both CMCs has influences on the DS value. This could be explained by the higher amount of carboxymethyl groups of the CMC having higher DS. This could be explained by the higher amount of carboxymethyl groups in CMCs with a larger DS. As we knew, the carboxymethyl group is hydrophilic. Thus, the higher the DS, the more immobilized water achieved in the system [9,23].

### 3.3. Morphology of Cellulose and CMC

The morphology of raw material, extracted cellulose, and CMC from palm bunch and bagasse was evaluated. Figure 4a and Figure 5a represent the SEM micrograph of the lignocellulosic material of palm bunch and bagasse, respectively. The fiber diameter of palm bunch and bagasse was around 80 and 120 μm, respectively. The fiber was composed of cellulose microfibrils linked with hemicellulose and lignin [42]. Before the extraction of the cellulose, the fiber surface had some substances covering it, which could be lignin and hemicellulose, encrusting the cellulose within [43]. The SEM images of the palm bunch (Figure 4a) and bagasse (Figure 5a) show that the substances covering palm bunch fiber seem to be higher than for that of bagasse, while the surface of the bagasse is clearly smoother than that of the palm bunch in these figures. This result is in agreement with the cellulose and lignin content of palm bunch and bagasse previously described in this paper. After extraction and bleaching treatments with various H_2_O_2_ concentrations (0%–40%), the SEM image of palm bunch cellulose indicated the removal of the encrusting substances from the surfaces of cellulose fiber (Figure 4b–f). Meanwhile, the SEM image of the bagasse cellulosic surface did not show very clear evidence of the removal of those substances (Figure 5b–f). This may be because the bagasse had a low amount of lignin at around 0.5× compared with that of the palm bunch. However, the approximate lignin level of the cellulose could be calculated using the Kappa number, and it was previously provided in Section 3.1. The results in that section showed that the lignin content of cellulose from both palm bunch and bagasse decreased after bleaching. Figure 4e,f and Figure 5e,f noticeably indicate cellulose degradation at too low pH levels (at 30%–40% H_2_O_2_) [44]. This observation can be seen as evidence to confirm the character of the cellulose mentioned previously. In Figure 4g–k, each CMCp has a smoother surface compared to its maternal material, cellulose. This smooth surface may be attributed to the chemical modification of the cellulose when converting it to CMC. Figure 5g–k shows that the surface of each CMCb was found to have no residue of encrusting substances. In addition, these CMCs show distorted shapes while their cellulose shows a fiber-like form, except cellulose-b-40. This shape alteration in the CMC may be attributed to chemical modification via CMC synthesis, which was probably found in the CMC synthesized from the cellulose containing low lignin content (<2.7% in this study). It was also similar to the morphology result found for sugarcane bagasse by Motaung and Mokhothu [45].

### 3.4. FTIR Spectroscopy of Cellulose and CMC

Because of the superior physical properties of the cellulose and CMC from 20% H_2_O_2_ bleaching, they were selected to determine the evidence of bleaching and carboxymethylation. The FTIR spectrum of unbleached cellulose of palm bunch (cellulose-p-0) is represented in Figure 6a. The broad band at 3350.3 cm^−1^ can be attributed to O–H stretching, the peak at 2904.8 cm^−1^ is due to C–H stretching, and the peak at 1024.2 cm^−1^ is ascribed to –O– stretching [9]. Bands between 1600 and 1370 cm^−1^ result from lignin, those between 1593.2, 1506.4, and 1421.5 cm^−1^ from aromatic phenylpropane skeleton vibrations, and those at 1369.4 cm^−1^ from phenolic hydroxyl groups [46]. For the bleached cellulose from palm bunch (cellulose-p-20), the spectra were similar to the cellulose-p-0. However, the absorbance of the bands between 1600 and 1370 cm^−1^ seemed to decrease indicating the decrease of lignin content in the cellulose structure. 

The FTIR spectrum of unbleached cellulose of bagasse (cellulose-b-0) is represented in Figure 6b. The broad band at 3336.8 cm^−1^ is due to O–H stretching. The peaks at 2891.3 and 1022.6 cm^−1^ are due to C–H stretching and –O– stretching, respectively. The peak at 1641.4 cm^−1^ results from the O–H bending of water [8,24]. The bands at 1421.5 and 1367.5 cm^−1^ can be attributed to the aromatic phenylpropane skeleton vibrations and phenolic hydroxyl groups, respectively, of lignin. When comparing the spectra of cellulose-b-0 with the bleached cellulose from bagasse (cellulose-b-20), the result is in agreement with the result found in palm bunch. The slight decrease in the absorbance of the bands of lignin confirms the decreasing lignin content after bleaching. 

For CMCp and CMCb, the FTIR spectrum shows that the frequency of the absorption bands was similar to those of their respective cellulose. This implies that cellulose and CMC have similar functional groups (Figure 6a,b). Additionally, the peaks at 1325.1 cm^−1^ for CMCp-20 and at 1323.2 cm^−1^ for CMCb-20 can be attributed to –OH bending vibration [20]. The absorption bands at 1591.3 and 1419.6 cm^−1^ for CMCp-20, and at 1587.4 and 1413.8 cm^−1^ for CMCb-20, were respectively due to antisymmetric and symmetric stretching vibrations of the COO– group [10,24]. This indicates the substitution of carboxymethyl groups on the cellulose structure. 

### 3.5. Characterization of CMC Films

In this section, CMC film from each type of CMCp and CMCb was produced, and its properties were investigated. The thickness of CMCp and CMCb films ranged between 0.127 and 0.213 mm, and between 0.081 and 0.111 mm, respectively. Table 2 lists the *L** and WI values, mechanical properties, solubility, and water barrier of these CMC films. The *L** and WI values of the films related to those of the CMCs. It was found that H_2_O_2_ concentration affected the tensile strength (TS), elongation at break (EB), solubility, and water vapor transmission rate (WVTR) of CMCp and CMCb films. Interestingly, the results seemed to correlate with the DS previously described in this document. Bleaching cellulose with 20% H_2_O_2_ appeared to yield the highest values of TS, solubility and WVTR for both CMC films. In terms of TS, it could be explained by the fact that the more carboxymethyl groups CMC occupied, the more intermolecular forces within the polymer chains resulted in an enhanced TS of the film [23]. The difference in strength between the CMCp and CMCb films depends on the relative content of their cellulose and lignin [42]. The EB of the film relates to its flexibility. In the bleaching process and CMC synthesis, the crystallinity of cellulose structure is altered and decreased [9,10]. This decrease in the crystalline region affects the increasing flexibility of the CMC film. For CMCp, when H_2_O_2_ was above 20%, the EB was likely to decline gradually. This may be because cellulose degrades with higher acid levels before converting to CMC. Nevertheless, the EB of CMCb was not significantly different when the H_2_O_2_ varied from 10% to 40%. This is because crystallinity may not be altered for the group of the CMCb. In the meantime, our solubility and WVTR results are in agreement with the results for CMC from bagasse pulp [47] and CMC from durian rind [9]. The rising DS value allowed the higher solubility and WVTR of the CMC films. This may be because of the hydrophilicity of carboxymethyl groups. 

## 4. Conclusions

Cellulose extracted from palm bunch and bagasse could be bleached with various concentrations of hydrogen peroxide (H_2_O_2_). Moreover, the products after bleaching could be utilized as the main raw material in the synthesis of carboxymethyl cellulose (CMC) by etherification between the cellulose and chloroacetic acid under alkaline conditions. The results showed that to receive CMC with a high yield, degree of substation, and viscosity, the H_2_O_2_ concentration should be should be properly operated. To obtain a CMC film with high mechanical properties (tensile strength and elongation at break), solubility, and water vapor transmission rate, the H_2_O_2_ concentration should be controlled as well. It was also demonstrated that, in this work, bleaching cellulose with 20% H_2_O_2_ resulted in the synthesized CMC products and films with the best physical properties.

## Figures and Tables

**Figure 1 polymers-12-01505-f001:**
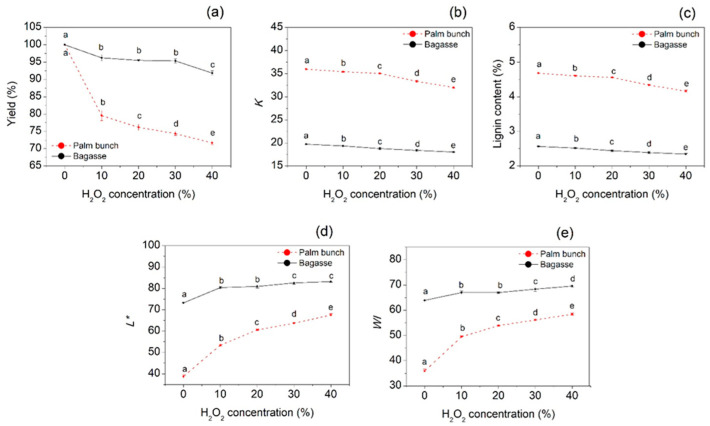
(**a**) Yield, (**b**) Kappa number, (**c**) lignin content and color values, (**d**) *L**, and (**e**) WI of cellulose from palm bunch and bagasse. Abbreviations: *L**, lightness; WI, whiteness index.

**Figure 2 polymers-12-01505-f002:**
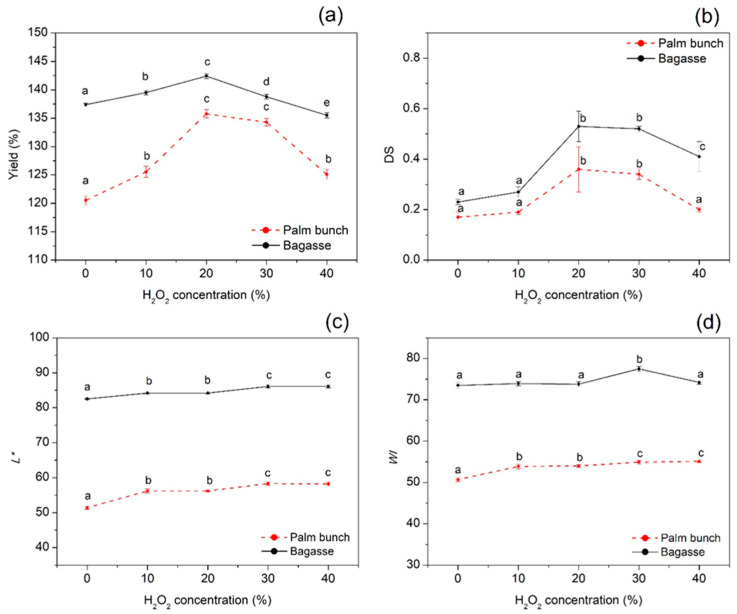
(**a**) Yield, (**b**) DS, (**c**) *L**, and (**d**) WI of CMC from palm bunch and bagasse. Abbreviations: DS, degree of substitution; CMC, carboxymethyl cellulose.

**Figure 3 polymers-12-01505-f003:**
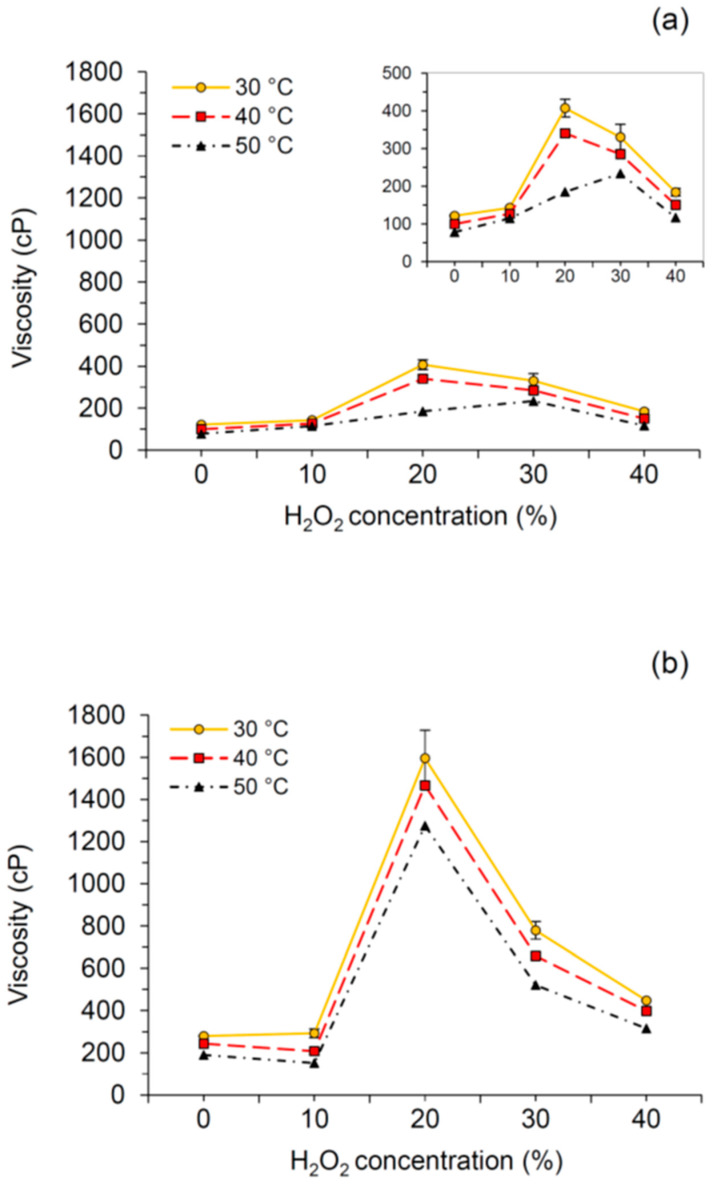
Viscosity of (**a**) CMCp and (**b**) CMCb at different temperature. Abbreviations: CMCp, CMC from palm bunch; CMCb, CMC from bagasse.

**Figure 4 polymers-12-01505-f004:**
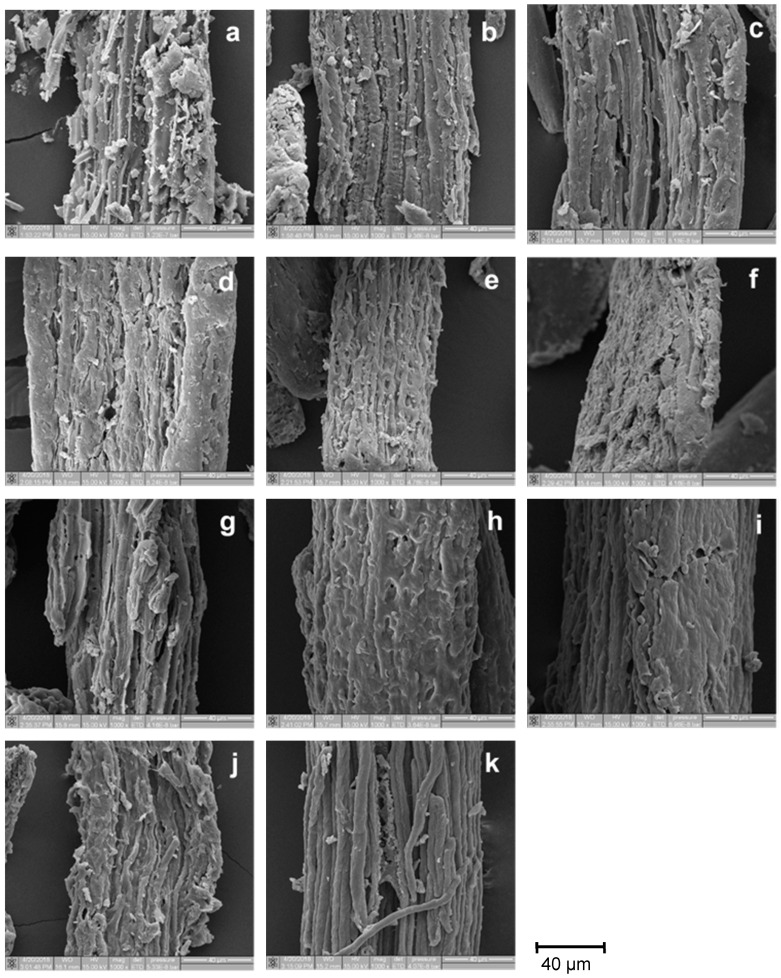
SEM images of (**a**) palm bunch, (**b**) cellulose-p-0, (**c**) cellulose-p-10, (**d**) cellulose-p-20, (**e**) cellulose-p-30, (**f**) cellulose-p-40, (**g**) CMCp-0, (**h**) CMCp-10, (**i**) CMCp-20, (**j**) CMCp-30, and (**k**) CMCp-40. Abbreviations: SEM, scanning electron microscope.

**Figure 5 polymers-12-01505-f005:**
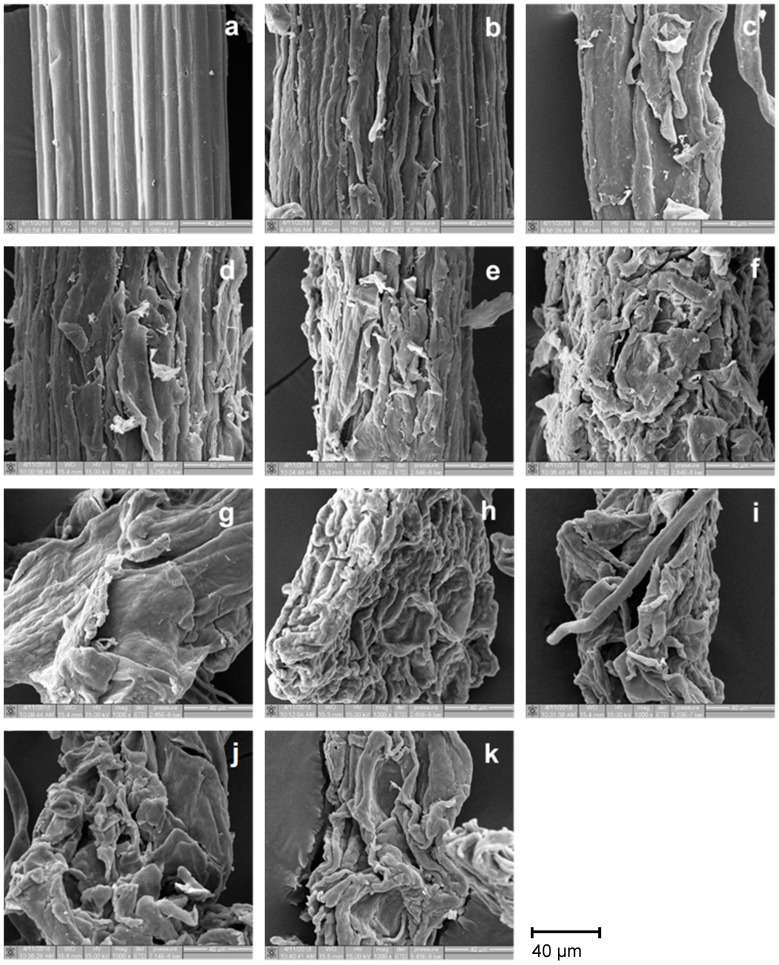
SEM images of (**a**) bagasse, (**b**) cellulose-b-0, (**c**) cellulose-b-10, (**d**) cellulose-b-20, (**e**) cellulose-b-30, (**f**) cellulose-b-40, (**g**) CMCb-0, (**h**) CMCb-10, (**i**) CMCb-20, (**j**) CMCb-30, and (**k**) CMCb-40.

**Figure 6 polymers-12-01505-f006:**
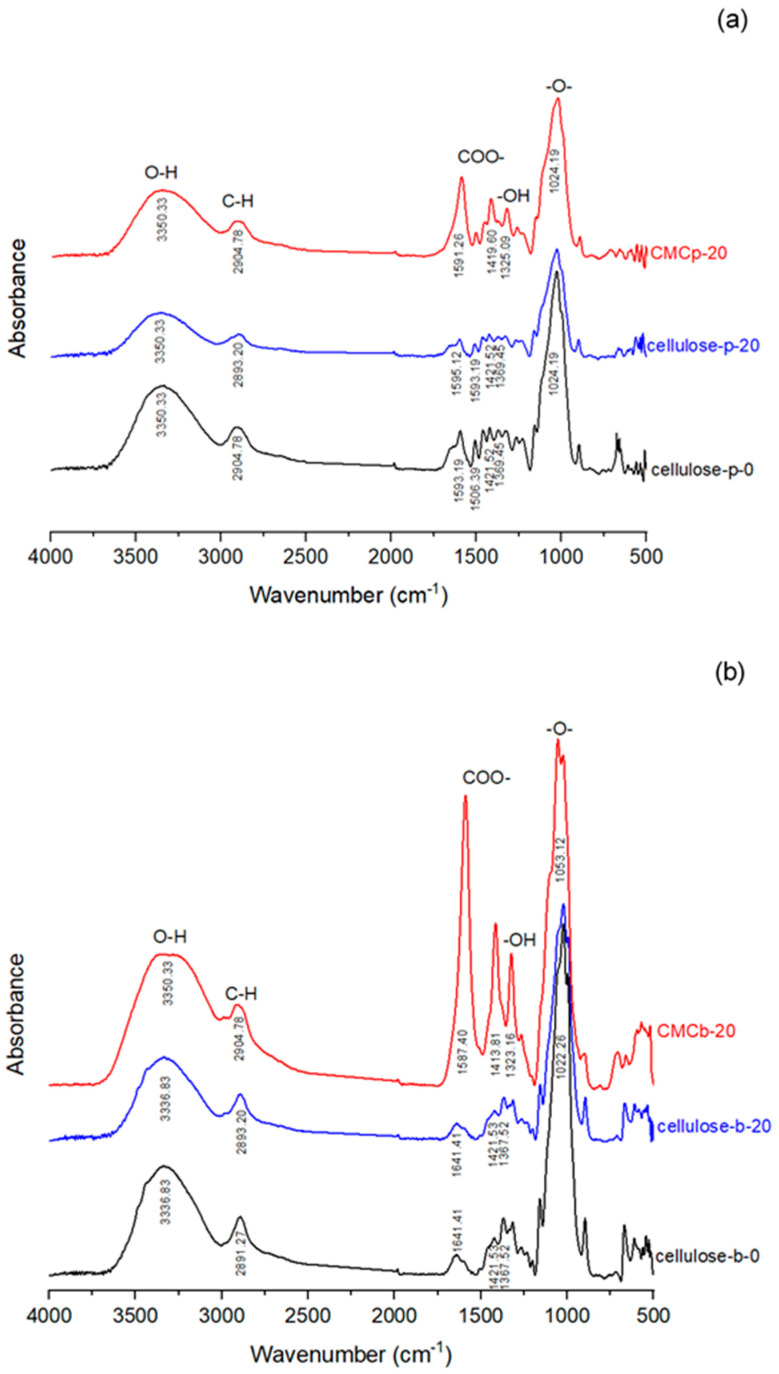
FTIR spectra of cellulose and CMC from (**a**) palm bunch and (**b**) bagasse.

**Table 1 polymers-12-01505-t001:** Appearance of cellulose and CMC from palm bunch and bagasse in different conditions.

Sample	Unbleached	Bleached with Various H_2_O_2_ Concentrations (*v*/*v*)			
	(0% H_2_O_2_)	10 %	20 %	30 %	40 %
Cellulose-p	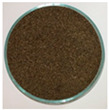	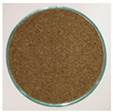	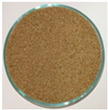	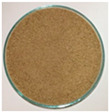	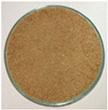
Code →	cellulose-p-0	cellulose-p-10	cellulose-p-20	cellulose-p-30	cellulose-p-40
Cellulose-b	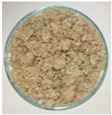	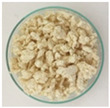	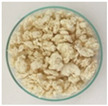	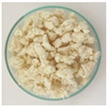	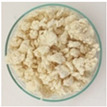
Code →	cellulose-b-0	cellulose-b-10	cellulose-b-20	cellulose-b-30	cellulose-b-40
CMCp	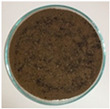	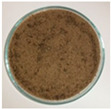	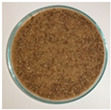	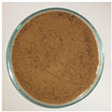	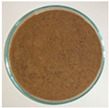
Code →	CMCp-0	CMCp-10	CMCp-20	CMCp-30	CMCp-40
CMCb	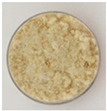	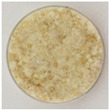	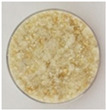	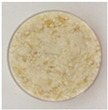	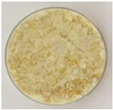
Code →	CMCb-0	CMCb-10	CMCb-20	CMCb-30	CMCb-40

Diameter of each container is 90 mm.

**Table 2 polymers-12-01505-t002:** Physical properties of the films from CMCp and CMCb.

Film	H_2_O_2_ (% *v*/*v*)	Thickness (mm)	*L**	WI	TS (MPa)	EB (%)	Solubility (%)	WVTR (g/d.m^2^)
CMCp	0	0.213 ± 0.052 ^a^	50.3 ± 0.3 ^a^	44.9 ± 0.5 ^a^	5.68 ± 1.67 ^a^	56.18 ± 26.17 ^ab^	28.44 ± 5.51 ^a^	58.15 ± 4.45 ^a^
	10	0.136 ± 0.006 ^bc^	60.2 ± 0.4 ^b^	47.6 ± 0.6 ^b^	7.77 ± 0.45 ^b^	68.73 ± 8.85 ^b^	34.85 ± 2.40 ^b^	58.23 ± 4.43 ^a^
	20	0.127 ± 0.011 ^b^	70.0 ± 0.7 ^d^	53.9 ± 0.9 ^c^	8.28 ± 1.06 ^b^	68.42 ± 7.57 ^b^	45.81 ± 1.36 ^c^	74.47 ± 8.93 ^b^
	30	0.144 ± 0.030 ^bc^	70.1 ± 0.4 ^d^	56.4 ± 0.6 ^d^	3.96 ± 1.14 ^c^	46.86 ± 14.44 ^a^	43.51 ± 3.78 ^c^	61.77 ± 7.24 ^a^
	40	0.156 ± 0.030 ^c^	68.9 ± 0.4 ^c^	57.1 ± 0.7 ^d^	1.41 ± 0.68 ^d^	42.86 ± 5.78 a	36.70 ± 1.91 ^b^	55.92 ± 7.23 ^a^
CMCb	0	0.111 ± 0.011 ^a^	85.3 ± 0.2 ^a^	81.5 ± 0.4 ^a^	13.94 ± 1.10 ^a^	79.08 ± 12.59 ^a^	36.12 ± 2.58 ^a^	46.28 ± 0.74 ^a^
	10	0.083 ± 0.010 ^b^	86.2 ± 0.2 ^b^	84.6 ± 0.4 ^b^	16.91 ± 4.11 ^b^	86.64 ± 22.67 ^b^	37.70 ± 1.06 ^a^	51.72 ± 4.64 ^ab^
	20	0.081 ± 0.007 ^b^	86.2 ± 0.2 ^b^	84.1 ± 0.4 ^b^	29.09 ± 3.35 ^c^	84.92 ± 7.74 ^b^	66.65 ± 1.94 ^b^	62.94 ± 11.97 ^b^
	30	0.089 ± 0.014 ^b^	86.3 ± 0.4 ^b^	84.5 ± 0.3 ^b^	17.96 ± 2.68 ^b^	84.18 ± 11.10 ^b^	65.94 ± 3.99 ^b^	48.63 ± 11.13 ^ab^
	40	0.088 ± 0.008 ^b^	86.6 ± 0.1 ^b^	85.4 ± 0.8 ^c^	16.86 ± 3.03 ^b^	84.20 ± 11.41 ^b^	44.19 ± 1.28 ^c^	45.74 ± 1.94 ^a^

Values within a column in the same group followed by the same letter are not significantly different (*p* > 0.05).

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
