# Peer review of "Physical Properties of Carboxymethyl Cellulose from Palm Bunch and Bagasse Agricultural Wastes: Effect of Delignification with Hydrogen Peroxide"

_polymers, 2020, doi:10.3390/polym12071505_

Round 1

Reviewer 1 Report

- Introduction: Are there any other studies devoted to the influence of oxidizing agent concentration on the final product properties but devoted to other cellulose sources.

- describe shortly methods of determination of cellulose and lignin content (paragraph 2.2) as well as all characterization procedures (WI, L*). Details and additional equations can be added as supplementary materials.

- if there is also lignin in a dry bleaching product, then Eq.1 does not correspond to cellulose yield

- lines 188-195: font

- is it possible that viscosity of the different cellulose and CMC is related to the solubility? Where all samples tested in viscosity analysis dissolved in water?

- bands position (Fig. 6) should be given without decimal places.

Author Response

Answer to the editor and reviewers for manuscript ID: polymers-826634, titled “Physical properties of carboxymethyl cellulose from palm bunch and bagasse agricultural wastes: effect of delignification with hydrogen peroxide," by Rungsiri Suriyatem, Nichaya Noikang, Tamolwan Kankam, Kittisak Jantanasakulwong, Noppol Leksawasdi, Yuthana Phimolsiripol, Chayatip Insomphun, Phisit Seesuriyachan, Thanongsak Chaiyaso, Pensak Jantrawut, Sarana Rose Sommano, Thi Minh Phuong Ngo, Pornchai Rachtanapun *

The authors thank the editor and the external reviewers for reviewing this manuscript and for their comments and suggestions, which greatly improved the quality of the paper. Any revisions in the main manuscript are highlighted using the "Track Changes". Detailed responses to the comments are provided below.

Reviewer 1

English language and style

( ) Extensive editing of English language and style required 
( ) Moderate English changes required 
(x) English language and style are fine/minor spell check required 
( ) I don't feel qualified to judge about the English language and style 

Yes

Can be improved

Must be improved

Not applicable

Does the introduction provide sufficient background and include all relevant references?

( )

(x)

( )

( )

Is the research design appropriate?

(x)

( )

( )

( )

Are the methods adequately described?

( )

(x)

( )

( )

Are the results clearly presented?

(x)

( )

( )

( )

Are the conclusions supported by the results?

(x)

( )

( )

( )

Comments and Suggestions for Authors

The authors thank the reviewer for taking the time to review the manuscript. We also thank the reviewer for the positive feedback.

Question 1: - Introduction: Are there any other studies devoted to the influence of oxidizing agent concentration on the final product properties but devoted to other cellulose sources.

Answer: The information was included in the introduction part. It now reads on page 2, lines 55-59,Different cellulose sources have been reported to implement H2O2 in the bleaching treatment such as papaya peel [12], durian rind [9], oil palm empty fruit bunch [13] and cotton [14]. In addition, some studies report the influence of oxidizing agent concentration on the properties of final products of various cellulose sources such as pure cellulose and paper [15], olive tree branch [16] and rice husk [17].”

Question 2: - describe shortly methods of determination of cellulose and lignin content (paragraph 2.2) as well as all characterization procedures (WI, L*). Details and additional equations can be added as supplementary materials.

Answer: We thank the reviewer for the comment. Shortly methods of cellulose content, lignin content, Kappa number and L* value was rewritten and modified. They now read on page 2, lines 90-99, “Shortly, sample powder (3 g), distilled-water (160 mL), glacial acetic acid (0.5 mL) and sodium chloride (1.5 g) were mixed in a beaker and stirred continuously for 1 h at 80 °C. The beaker was cooled to 10 °C, filtered, and washed with acetone. The product was dried in an oven for 24 h at 80 °C. Cellulose content (%) was calculated by multiplying the ratio of product weight/sample weight with 100. In order to determine lignin content, briefly, sample powder (1 g) was placed in a beaker at 2 °C. Cold (10-15 °C) sulfuric acid (72%, 15 mL) was gradually introduced, and the mixture was continuously stirred for 2 h. Distilled-water (560 mL) was added. The mixture was boiled for 4 h and left standing to allow precipitation overnight. The mixture was filtered, washed with water three times, and dried in an oven for 6 h at 105 °C. The final product was later weighed. Lignin content (%) could be calculated by multiplying the ratio of product weight/sample weight with 100.”

, page 3, lines 114-124, “Briefly, the pulp (3 g) was suspended in distilled-water (500 mL) with continuous stirring for 5 min at 25 °C. The addition of 0.1 M potassium permanganate (100 mL) and 4 M sulfuric acid (100 mL) were later introduced, and the mixture was then stirred continuously for 5 min. Distilled-water was added to achieve the total volume of 1.0 =L with constant stirring for 10 min. The cessation of reaction was resulted with addition of 1 M potassium iodide (20 mL). Immediately, the free iodine in the mixture was titrated with 0.2 M sodium thiosulfate. A few drops of starch solution were used as an indicator. K value could be calculated using the equation described elsewhere [27]. Approximate lignin level (%) of the bleached cellulose samples was calculated by using Eq. 2 [27] with:

                                                         (2)

, and page 3, lines 144-148, “Lightness (L*) value of the cellulose and CMC samples was measured by using a colorimeter (CR-400, Konica Minolta, China). Sample powder (2 g) was sit in a sample holder placing above the instrument. The testing button was pressed, and L* value was shown on the result screen. Each test was run in triplicate. Whiteness index (WI) value was calculated by using the method described elsewhere by the authors [28].” 

Question 3: - if there is also lignin in a dry bleaching product, then Eq.1 does not correspond to cellulose yield

Answer: We thank the reviewer for the good comment. We agree with reviewer that bleaching products still have some lignin left in the products. We can see from Figure 1b. Palm bunch fiber has higher Kappa Number higher than bagasse fiber. It means both beached fibers still have lignin left. However, it is called “Yield of cellulose” because lignin content is beached cellulose is quite low compared with the whole cellulose. From the literature, this is the equation which uses to calculate the yield of cellulose.

J Chumee, D Seeburin (2014) Cellulose extraction from pomelo peel: Synthesis of carboxymethyl cellulose - International Journal of Materials and Metallurgical Engineering, 8 (5), 435-437.

Pornchai Rachtanapun, Suwaporn Luangkamin, Krittika Tanprasert and Rungsiri Suriyatem (2012) Carboxymethyl Cellulose Film from Durian Rind, LWT-Food Science and Technology, 48, 25-28.

Question 4: - lines 188-195: font

Answer: The whole paragraph was checked and corrected.

Question 5: - is it possible that viscosity of the different cellulose and CMC is related to the solubility? Where all samples tested in viscosity analysis dissolved in water?

Answer: In viscosity analysis, all samples were tested in water. It is possible that viscosity is related with DS value and solubility of the CMC as well. As we mentioned in section 3.2 on page 8, line 250-252, that “The DS was used to indicate the solubility of the CMC. The DS value between 0.0-0.4 means CMC is insoluble but swellable, and above this range means the CMC is completely water-soluble.”. To the best of our knowledge, for our CMCs, the more DS, the more water soluble and the more viscous. However, the limitation of this possibility may be further studied in the next project. 

Question 6: - bands position (Fig. 6) should be given without decimal places.

Answer: We thank the reviewer’s suggestion. However, we respectfully disagree. So, we will prefer to keep it. If the editor thinks that the bands position should be given without decimal places, we will do it.

Reviewer 2 Report

All my comments are listed below with an appropriate Line number(s) in order to facilitate tracking:

Line 28: I think that it should be "to synthesize" instead of "to synthesis"?

Line 38: Add "the" in front of "lowest".

Line 45: Delete "in". It is surplus here.

Line 47: Suggest to rewrite "the cell walls of many plants" as "the plant's cell walls". It is simplier and more readable.

Line 49: Replace "on with "and" since there is no such thing as "composition of celulose content".

Line 52: Suggest to replace "very few" with "just a" as well as "types environmentally safe" with "eco-friendly reagents" or some similar construction.

Line 55: "eco-materials" instead of "environmentally material".

Lines 55-56: Suggest to authors to replace "and tend to be on interest further on" with "...with constant growing interest". It sounds to me more logical and appropriate here.

Line 80 Add "pieces of" in front of "20 mm".

Line 80: Is something missing in "length-sized.... two days"? Please check and add/correct if it is necessary.

Line 98: Suggest to replace "." with ":" after "Eq. 2 (22)" since it will continue with equation 2 in the next line.

Line 147: The same as previous.

Line 164: Delete "by Coral Medin et al.". It is surplus here.

Lines 188-195: There is technical issue with this part of text since it is not in line with text style applied through a whole Manuscript. Please, check and correct adequately.

Line 203: Suggest to authors to start sentence as: "At the beginning percent yield..." It seems to me more logical since you did not have a constant yield increase during all applied treatments.

Line 206: I think that it can not be "mother material" but "maternal material" or some similar phrases.

Line 248: Can you plase explain in which way you think that it is higher for palm bunch fiber. I do not understand completely.

Lines 251-252: On Fig. 5 "removal of the encrusting substances" is not quite "visible" as it is on Figure 4. In my oppinion, Fig. 5 is pretty much debatable.

Line 255: I must say that I can not see the same as you on Figures 5g-k on the contrary to Figures 4g-k where it is quite clear and concise. So, I must ask to authors to make some additional effort to clarify and explain differences between Figures 4 and 5.

Line 273: Suggest to authors to replace "owing" with "ascribed". It seems more logical to me in this context.

Lines 301-302: Please add unit for numerical values.

Author Response

Response to Reviewer 2

Answer to the editor and reviewers for manuscript ID: polymers-826634, titled “Physical properties of carboxymethyl cellulose from palm bunch and bagasse agricultural wastes: effect of delignification with hydrogen peroxide," by Rungsiri Suriyatem, Nichaya Noikang, Tamolwan Kankam, Kittisak Jantanasakulwong, Noppol Leksawasdi, Yuthana Phimolsiripol, Chayatip Insomphun, Phisit Seesuriyachan, Thanongsak Chaiyaso, Pensak Jantrawut, Sarana Rose Somman, Thi Minh Phuong Ngo, Pornchai Rachtanapun *

The authors thank the editor and the external reviewers for reviewing this manuscript and for their comments and suggestions, which greatly improved the quality of the paper. Any revisions in the main manuscript are highlighted using the "Track Changes". Detailed responses to the comments are provided below.

Reviewer 2

Open Review

English language and style

( ) Extensive editing of English language and style required 
( ) Moderate English changes required 
(x) English language and style are fine/minor spell check required 
( ) I don't feel qualified to judge about the English language and style 

Yes

Can be improved

Must be improved

Not applicable

Does the introduction provide sufficient background and include all relevant references?

(x)

( )

( )

( )

Is the research design appropriate?

(x)

( )

( )

( )

Are the methods adequately described?

(x)

( )

( )

( )

Are the results clearly presented?

( )

(x)

( )

( )

Are the conclusions supported by the results?

(x)

( )

( )

( )

Comments and Suggestions for Authors

Question 1: All my comments are listed below with an appropriate Line number(s) in order to facilitate tracking:

Answer: The authors thank the reviewer for taking the time to review the manuscript. We also thank the reviewer for the positive feedback.

Question 2: Line 28: I think that it should be "to synthesize" instead of "to synthesis"?

Answer: The word was changed as suggested.

Question 3: Line 38: Add "the" in front of "lowest".

Answer: The word was added as suggested.

Question 4: Line 45: Delete "in". It is surplus here.

Answer: The word was deleted as recommended.

Question 5: Line 47: Suggest to rewrite "the cell walls of many plants" as "the plant's cell walls". It is simplier and more readable.

Answer: The phrase was rewritten as requested.

Question 6: Line 49: Replace "on” with "and" since there is no such thing as "composition of celulose content".

Answer:  The word was changed as suggested.

Question 7: Line 52: Suggest to replace "very few" with "just a" as well as "types environmentally safe" with "eco-friendly reagents" or some similar construction.

Answer: They were changed as suggested.

Question 8: Line 55: "eco-materials" instead of "environmentally material".

Answer: The word was changed as recommended.

Question 9: Lines 55-56: Suggest to authors to replace "and tend to be on interest further on" with "...with constant growing interest". It sounds to me more logical and appropriate here.

Answer: The phrase was changed as suggested.

Question 10: Line 80 Add "pieces of" in front of "20 mm".

Answer: The word was added as suggested.

Question 11: Line 80: Is something missing in "length-sized.... two days"? Please check and add/correct if it is necessary.

Answer: The words were checked and corrected.

Question 12: Line 98: Suggest to replace "." with ":" after "Eq. 2 (22)" since it will continue with equation 2 in the next line.

Answer: The word was added as suggested.

Question 13: Line 147: The same as previous.

Answer: The word was added as suggested.

Question 14: Line 164: Delete "by Coral Medin et al.". It is surplus here.

Answer: The words were deleted as requested.

Question 15: Lines 188-195: There is technical issue with this part of text since it is not in line with text style applied through a whole Manuscript. Please, check and correct adequately.

Answer: The whole paragraph was checked and corrected.

Question 16: Line 203: Suggest to authors to start sentence as: "At the beginning percent yield..." It seems to me more logical since you did not have a constant yield increase during all applied treatments.

Answer: The words were changed as suggested.

Question 17: Line 206: I think that it cannot be "mother material" but "maternal material" or some similar phrases.

Answer: The word was changed as recommended.

Question 18: Line 248: Can you please explain in which way you think that it is higher for palm bunch fiber. I do not understand completely.

Answer: The sentences were modified. It now reads on page 10, lines 288-291, “The SEM images of palm bunch (Figure 4a) and bagasse (Figure 5a) showed that the covering substances of palm bunch fiber seemed to be higher than that of bagasse.” While the surface of bagasse was clearly smoother than that of palm bunch at these figures. 

Question 19: Lines 251-252: On Fig. 5 "removal of the encrusting substances" is not quite "visible" as it is on Figure 4. In my opinion, Fig. 5 is pretty much debatable.

Answer: The sentences were modified. It now reads on page 10, lines 286-294, After the extraction and bleaching treatments with various H2O2 concentrations (0-40%), the SEM image of palm bunch celluloses indicated the removal of those encrusting substances on the surfaces of cellulose fiber (Figure 4b-f). Meanwhile, the SEM image of the bagasse cellulosic surface did not show very clear evidence of the removal of those substances (Figure 5b-f). This maybe because bagasse had low amount of lignin as around 0.5x comparing with that of palm bunch. However, approximate lignin level of the celluloses could be calculated via Kappa number and it was previously provided in section 3.1. The results in that section showed that the lignin content of both celluloses from palm bunch and bagasse decreased after bleaching.”

Question 20: Line 255: I must say that I cannot see the same as you on Figures 5g-k on the contrary to Figures 4g-k where it is quite clear and concise. So, I must ask to authors to make some additional effort to clarify and explain differences between Figures 4 and 5.

Answer: We thank the reviewer’s comment. We understand the request of the reviewer about this discussion. We have modified the sentences to better explain this point. It now reads on page 10, lines 340-310, “In Figure 4g-k, each CMCp had a smoother surface when it was compared to its maternal material, cellulose. This smooth surface may be attributed to chemical modification of the cellulose converting to CMC. In Figure 5g-k, the surface without residue of encrusting substances was found for each CMCb. In addition, these CMCs showed distorted shapes while their celluloses showed fiber like-form, except the cellulose-b-40. This shape alteration in the CMC may be attributed to the chemical modification via CMC synthesis, which was probably found in the CMC synthesized from the cellulose containing low lignin content (< 2.7% in this study).”

Question 21: Line 273: Suggest to authors to replace "owing" with "ascribed". It seems more logical to me in this context.

Answer: The word was changed as suggested.

Question 22: Lines 301-302: Please add unit for numerical values.

Answer: The unit was added as requested.
